# Dissimilar Materials Welding with a Standoff-Free Vaporizing Foil Actuator between TRIP 1180 Steel Sheets and AA5052 Alloy

**DOI:** 10.3390/ma14174969

**Published:** 2021-08-31

**Authors:** Yuhyeong Jeong, Giseung Shin, Choo Woong, Jeoung Han Kim, Jonghun Yoon

**Affiliations:** 1Department of Mechanical Engineering, Hanyang University, Seoul 04763, Korea; jtpye9402@gmail.com; 2BK21 FOUR ERICA-ACE Center, Hanyang University, Gyeonggi-do, Ansan 15588, Korea; 3Department of Materials Science & Engineering, Hanbat National University, Daejeon 34158, Korea; rltmd07@gmail.com (G.S.); cu9767@naver.com (C.W.); jh.kim@hanbat.ac.kr (J.H.K.); 4Department of Mechanical Engineering, Hanyang University ERICA, Gyeonggi-do, Ansan 15588, Korea

**Keywords:** vaporizing foil actuator welding, standoff-free, dissimilar materials welding, impact welding, lap shear test, metallic bonding

## Abstract

This paper mainly demonstrates an advanced type of the vaporizing foil actuator welding (VFAW) process between GPa-grade steel (TRIP1180) and aluminum alloy (AA5052-H32) without applying standoff. To secure a flying distance during the VFAW process, the preformed target sheet shaped like a circular indentation has been utilized. It is necessary to optimize process parameters integrated with geometrical design of the preform since the welding strength can be decreased beyond the optimum input energy in the standoff-free VFAW process. The welded surface was evaluated by SEM-EDS, XRD, EBDS, and TEM to analyze the welding mechanism and composition at the welding interface. The diffusion zone including the AlFe_3_ phase was observed at the welded interface which has high grain density due to the high-speed impact by increasing the welding strength, which leads to the perfect welding between the dissimilar materials.

## 1. Introduction

There is an increasing demand for a lightweight design in the body structure of the transportation vehicles, especially for electric and hybrid cars to achieve high energy efficiency [1,2] and decrease gas emissions [3,4]. It is the one of the best strategies for lightening body weight with joining and welding different grades of materials such as GPa grade steel and Al alloy with each other, which makes it possible to have a strength gradient in the single panel. In order to weld dissimilar materials, different types of welding processes such as fusion and solid-state welding [5] have been developed depending on whether it involves melting and subsequent solidification or does not. Even though fusion welding such as arc welding [6,7], gas welding [8,9], and power beam welding [10,11] has been widely applied to various industries, there are several issues related with defect formation around a welded interface since they tend to induce local melting with phase transition at the interface during the welding process in which large variation in mechanical and thermal properties such as strength, elongation, and thermal expansion ratio, etc. are able to cause residual stress along the interface.

The solid-state welding applies a sufficient external force or pressure to induce plastic deformation at the interface, which can be divided into hot pressure welding including resistance spot welding [12,13] and cold pressure welding such as friction stir welding [14,15,16,17] explosion welding [18], and self-piercing riveting (SPR) [19,20] according to the heat generation at the interface to facilitate bonding. Although solid-state welding has been considered as a potential method due to its reduced effect on the thermal fractures, the welding strength is subjected to be influenced by the quality of welding interface at the preparation stage in terms of the amount of the oxide layer or surface cleaning with degreasing or brushing, etc. [19]. Additionally, the excessive physical deformation at the welded sheets can cause the fracture or delamination at the interface [20,21].

The vaporizing foil actuator welding (VFAW) is the one of the explosion welding, which does not involve the conventional full melting of the materials being joined. It applies substantially high pressure to the bottom of a flyer sheet as depicted in Figure 1a to make it collide with a target sheet, which tends to induce metallic bonding [22,23,24] at the interface for permanent welding. When a high current is instantly applied to the aluminum foil, it is vaporized from solid to gas, directly, which tends to generate tremendously high pressure as shown in Figure 1b. The generated explosive pressure sharply pushes the flyer sheet with substantially high velocity, and it distributes along the welding interface, which tends to induce an oblique angle between the target and flyer sheets as shown in Figure 1c. This explosive pressure ejects oxides and other contaminants on the interface with leaving behind fresh metal surfaces, which leads to the formation of a metallic bond between the two sheets by attaching clean metallic surfaces [25,26].

Chen et al. [26] and Liu et al. [27,28] have demonstrated the VFAW process to weld dissimilar materials such as various grades of steel with light-weight alloys including aluminum and magnesium alloys. Vivek et al. [29,30] successfully carried out the welding between the titanium–copper alloy and the BMG (Bulk Metallic Glass)-Cu110 alloy in which they observed the morphology of welded interface with respect to the impact velocity of the flyer sheet. Liu et al. [22] have investigated Al_x_Fe_y_ phases at the welded interface with supported by EDS-SEM, EBSD, and TEM analyses. It has also been researched to optimize the process parameters such as impact angle, standoff distance, and input energies, etc. during the VFAW process. Vivek et al. [29] have examined the effect of impact angle and velocity on weldability by controlling the amount of input energy and standoff distance. However, the usage of standoff causes lots of problems from the practical aspect in the industrial process even though it tends to secure an allowable distance for the flyer sheet to guarantee the desired impact speed. It does not only increase the weight of the welded part due to redundant additional standoff, which has nothing to do with the two target materials, but also requires an undesirable secondary process to eliminate this part if it is necessary. In addition, since a slight misalignment between target, flyer sheets, and standoff at the initial set-up when stacking up with each other is able to induce strength variation in the welding interface, it tends to require a high level of tolerance control in the process. Under these circumstances, it is required to eliminate the standoff during the VFAW process for enhancing manufacturing efficiency.

In this paper, we have proposed a standoff-free VFAW process without applying a conventional standoff that is replaced by a pre-deformed target sheet with support by a simple stamping process. To optimize the preform shape, FEM analysis has been carried out, which was confirmed by experiments for perfect welding between a TRIP1180 steel sheet and an AA5052 sheet. In order to validate the sufficient welding strength in terms of mechanical properties and metallurgical aspects, a lap shear test and microstructure investigation with SEM-EDS and TEM along the interface have been conducted.

## 2. Experimental Procedure

Figure 2 shows the initial set-up for the conventional VFAW process which consists of flyer sheet, standoff, and target sheet, which is sequentially stacked on the Al foil as depicted in Figure 2b. Figure 3 demonstrates the dimensional specification of the Al foil in which an actuating area is designed to be narrow for concentrating a high current to activate a local evaporation [23]. The actuating length of 2 mm has been applied to the experiment to increase the impact velocity since it tends to influence the amount of vaporizing pressure directly. To guarantee the uniform contact pressure in between specimens and experimental safety against explosive pressure during the VFAW test, the top and bottom surface of the stacked specimens are tightened by a back-up die set. In addition, Kapton film has been utilized to insulate the Al foil from the die set and the other welding specimens, which tends to prevent a current loss [23]. After an initial set-up, a target current from the capacitor bank directly flows to the Al foil through the copper plate as shown in Figure 4. The specifications of the capacitor bank for VFAW test are the maximum voltage of 8 kV, maximum energy of 12.8 kJ, and a capacitance of 200 μF.

To demonstrate practical welding between GPa grade steel and Al alloy, TRIP1180 and AA5052-H32 sheets with a length, width, and thickness of 100 mm × 50 mm × 1.2 mm, and 200 mm × 50 mm × 1.0 mm, respectively, have been applied to the VFAW process as a target and flyer sheet, respectively. The tensile tests for the initial mechanical properties were conducted with the ASTM-E8 standard [31] specimen as represented as Figure 5, in which they show the ultimate tensile strength of 1218.65 MPa and 221.51 MPa for TRIP1180 and AA5052-H32, respectively.

To validate the welding strength, a lap shear test has been carried out in the universal testing machine (UTM) with the cross-head speed was 0.1 mm/s. The welded TRIP1180 and AA5052 specimen were gripped at the UTM and pulled toward the test direction. Figure 6 demonstrates the schematic design of the lap shear test in which the welded interface coincides with the centerline of the test set-up for inducing pure shear deformation along the interface [25,29].

For the microscopic investigation at the welded interface, cross-sections of the welded surface were prepared, first, by mechanical grinding using a SiC girt paper (400 grit to 2400 grit (8 μm grain size)), then polished with suspensions of 1 and 0.25 μm diamond polishing solution, and finished with further polishing using 0.05 μm colloidal silica suspension to obtain the required surface finish. The sample surface after final polishing was cleaned under running water, followed by ethanol, and dried. After surface preparation, the microstructure and elemental composition of the joints were examined using scanning electron microscopy (SEM, JSM-5800 JEOL, Tokyo, Japan) coupled with energy dispersive spectroscopy (EDS) and electron backscatter diffraction (EBSD) operated at an accelerating voltage of 20 kV. The TSL OIM Analysis v8 software was used to process EBSD data and generate inverse pole figure (IPF), image quality (IQ), and phase maps. Microbeam X-ray diffraction (XRD) analysis was carried out using a Rigaku D/Max Rapid-S with CuK radiation to identify the phases in the narrow joints. Transition electron microscope (TEM) analyses were carried out on the cross sections of the joint interface using the Cs-corrected scanning transmission electron microscope (STEM, Hitachi-HF5000 instrument, Amsterdam, Netherland) operated at 200 kV. The specimen used in the TEM analysis was prepared by lifting the joint interface using a focused ion beam (FIB, FEI Helios Nano-Lab 600, Hillsboro, OR, USA).

## 3. Standoff-Free Vaporizing Foil Actuator Welding

### 3.1. Preform Design of the Target Sheet

It has been proposed to perform the VFAW test with the application of a preformed target sheet instead of utilizing a conventional standoff. To secure a sufficient flying distance, the TRIP1180 target sheet with a thickness of 1.2 mm has been stamped to have an indentation along the circular boundary as depicted in Figure 7, which is applied to the initial stacking for the standoff-free VFAW test. The preform die set is designed to impress a circular indentation [32] on the target sheet with a diameter and height of 30 mm and 1.6 mm since it has been validated in the conventional VFAW experiments [23,24,25,29,30] for guaranteeing the optimum welding area and flying distance. The design variables in the specification of the die set are represented in Figure 8. For the various combinations of design variables as shown in Table 1, the final dimensions for R1 and R2 have been selected as 1.5 mm, respectively, not to induce material failure during the stamping process. Figure 9 demonstrates the FE analysis results with ABAQUS/Standard [33], which represents uniform strain distribution without inducing strain localization around the corner when R1 and R2 have been selected as 1.5 mm. Figure 10 demonstrates the preform die set based on the optimal design variables, which is installed in a 100-tonf servo press for carrying out the preforming process. It is able to produce the desired preform shape without inducing material failure as depicted in Figure 10b.

### 3.2. Lap Shear Test for a Welded Specimen

With applying the preformed target sheet, the standoff-free VFAW test has been performed with respect to the increase of the input energy from 4 kJ to 12 kJ to examine the effect of the input energy on the welding strength between TRIP1180 and AA5052. Figure 11 shows the experimental results of lap shear test, which has been conducted until the final fracture occurs. Three experiments were conducted under the same input energy conditions for repeatability evaluation. Since the delamination occurs due to imperfect welding between the target and flyer sheets, it is not able to sustain sufficient reaction force compared with the flyer sheet made of AA50502 as shown in Figure 12 when the input energy of 4 and 6 kJ is applied. However, it is noted that the welded specimen with the input energy of 8 kJ shows an early fracture during the lap shear test in the vicinity of the welding zone as depicted in Figure 12 since it tends to induce undesirable thickness reduction around the corner of circular welded region even though it the delamination does not occur. When the input energy increases to 10 kJ, there is no other early fracture and delamination due to a perfect welding at the interface as depicted in Figure 11d, which results in the final fracture at the flyer sheet itself instead of the circular welded region as shown in Figure 12. To confirm the thickness distribution with respect to the input energy, the welded specimen is sectioned in half by waterjet cutting as shown in Figure 13. Since the thickness reduction has occurred in both of sheets with similar ratios such as 5–11.6% in the target sheet and 8–12% in the flyer sheet during the VFAW, the early fracture is attributed to the localization in the flyer sheet with relatively low strength at the *x*-coordinate of 16.5 mm. It is interesting to note that the thickness of the flyer sheet has only increased a lot in case of 10 kJ as depicted in Figure 14b while it decreased when the input energy of 6, 8, 12 kJ has been applied compared with the initial sheet thickness. This is why the welded specimen with applying the input energy of 10 kJ tends to exhibit comparable tensile strength to the AA5052 without showing delamination and early fracture at the interface. It is noteworthy that it is necessary to take into consideration optimum process parameters with correlating the amount of input energy, simultaneously, since the weldability between dissimilar materials from the VFAW test is not proportionally influenced by the amount of input energy only [24,26].

### 3.3. Microstructure Investigation in the Welding Interface

The welding interface between TRIP1180 and AA5052 sheets has been investigated with SEM-EDS, micro XRD, EBSD, and TEM. Four specimens were extracted from the perfect welded specimen for the microstructure investigation by wire cutting as shown in Figure 15 and polished using up to 4000 grit sandpaper.

There is a compositional diffusion zone between the TRIP1180 and AA5052 interface, which was generated by excessive plastic deformation by high-speed impact as depicted in Figure 16. Since there is no mechanical interlocking force due to the flat interface morphology, it appears that the metallic bonding is formed at the welding interface. The detailed investigation to find out metallic bonding at the diffusion zone was implemented using micro XRD. Figure 17 shows the results of the micro XRD measurements of the aluminum, steel, and interface. Even if the intensity peak at the interface is different with the steel and aluminum phase, it is difficult to confirm the formation of Al-Fe phase because the intensity peak of the Al-Fe phase is very close to those of the Al and Fe phases. EBSD measurements were conducted further to identify the potential phase formation at the interface. Figure 18 shows the results of the EBSD measurements at the interface where the potential Al-Fe phase was observed. In the image quality (IQ) mapping, very low confidential index was noted at the interface due to the very large plastic deformation which results in lattice distortion. In the inverse pole figure (IPF) image, a random crystallographic orientation of the constituent crystals was noticed. The phase map demonstrated that the interface zone mostly consists of the Al-Fe phase while the Al side was populated with FCC crystals and the Fe side with BCC crystals. In the kernel average misorientation (KAM), many misorientation levels at the interface were high compared with those of parent material regions. From these results, the new phase of Al-Fe was formed at the interface, which has a very dense grain structure because of the high-speed impact. In order to characterize the phase more precisely, TEM work has been performed around the interface region.

Figure 19a indicates the TEM observation area of the FIB sample. Figure 19b,d shows typical FFT patterns of Al and Fe. In contrast, an SAED pattern obtained at the interface revealed a ring pattern indicating the amorphous formation or a very fine nano-grain structure. The SAED ring pattern was well indexed to the AlFe_3_ with a space group of Fm$\bar{3}$m, which is thermodynamically stable [34,35]. Liu et al. [22,27] also observed complex intermetallic AlFe phases at the interface zone such as AlFe, AlFe_3_, Al_5_Fe_2_, and Al_3_Fe. Perhaps, meta-stable intermetallics such as Al_2_Fe with low symmetry would be transformed to a high-symmetry phase [36]. From these results, it seems that the driving force of the high weldability between the two materials resulted from the metallic bonding, as observed in the microstructural analysis.

When the flyer sheet collides with the target sheet with high speed and high pressure, the local temperature of the surface around the impact region should be high enough to form metallic bonding, eventually resulting in the formation of the AlFe_3_ intermetallic phase that is hard and brittle in nature. Generally, the existence of such kind of intermetallic phases is not preferable for enhanced joint strength and ductility. The crack found in Figure 18 indicates the brittleness around the region. However, even with this AlFe_3_ intermetallic phase, very high joint strength could be achieved due to metallic bonding between Al and Fe sheets.

## 4. Conclusions

In this paper, the VFAW welding of dissimilar materials between TRIP1180 and AA5052-H32 has been conducted by substituting a standoff with the preformed shape in the target sheet to increase the efficiency of the VFAW process. The design parameters of the preformed shape were optimized through the FEM analysis considering the restriction conditions from the geometrical limit of the preformed shape, which makes it possible to have a perfect welding between TRIP1180 and AA5052-H32 by applying the input energy of 10 kJ. It has been concluded that it is substantially necessary to optimize process parameters integrated with the geometrical design of the preform since the welding strength can be decreased beyond the specific input energy due to the nonlinearity of the process parameters in the standoff-free VFAW process. Many microstructural observations have been conducted to identify the composition and phase at the welding interface. From the SEM-EDS and micro XRD results, diffusion between the aluminum and steel was observed, but it was not confirmed that the new phase was formed at the interface. Like the results of the EBSD and TEM analysis, the AlFe_3_ phase was observed at the interface which has a very fine grain structure because of the high speed impact. It can be concluded that the metallic bonding occurred at the interface during the VFAW process by forming the AlFe_3_ phase, which results in high welding strength between the dissimilar materials.

## Figures and Tables

**Figure 1 materials-14-04969-f001:**
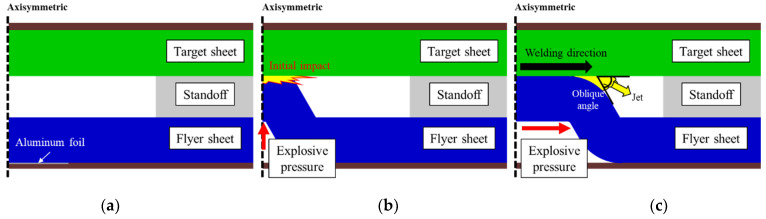
Welding mechanism of VFAW process: (**a**) initial set-up; (**b**) initial impact due to Al foil vaporization; (**c**) explosive pressure distribution.

**Figure 2 materials-14-04969-f002:**
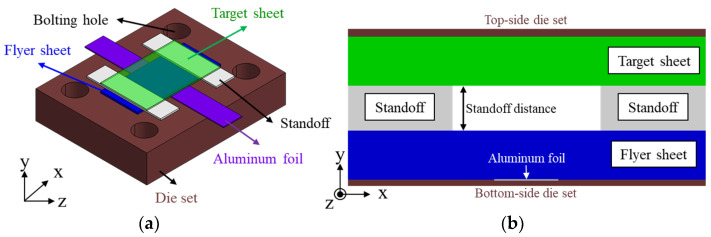
Initial set-up of the conventional VFAW experiment: (**a**) stacking sequence with specimen and vaporizing Al foil; (**b**) sectional view for stacked specimen.

**Figure 3 materials-14-04969-f003:**
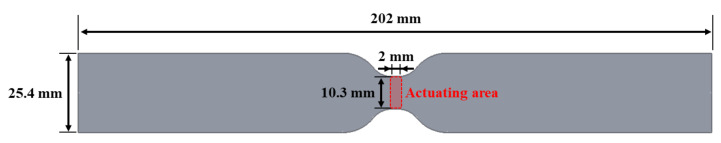
Specification of aluminum foil specimen.

**Figure 4 materials-14-04969-f004:**
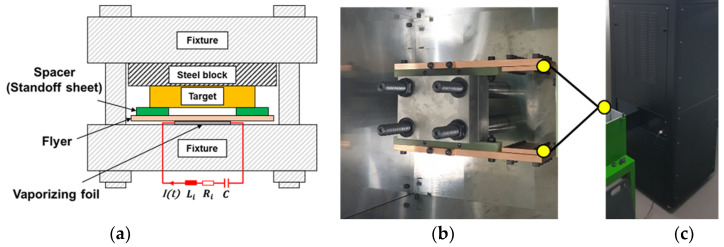
VFAW test equipment including die set and capacitor bank: (**a**) initial setting for VFAW experiments at the side view; (**b**) die set with copper plate at the top view; (**c**) capacitor bank.

**Figure 5 materials-14-04969-f005:**
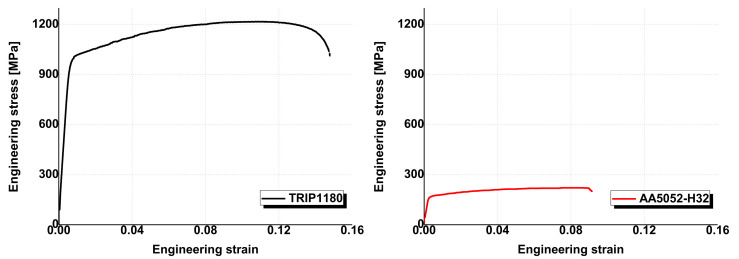
Stress–strain curves at the quasi-static state for TRIP1180 and AA5052.

**Figure 6 materials-14-04969-f006:**
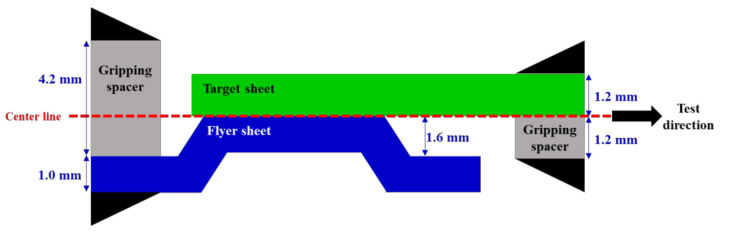
Schematic design of the lap shear test.

**Figure 7 materials-14-04969-f007:**
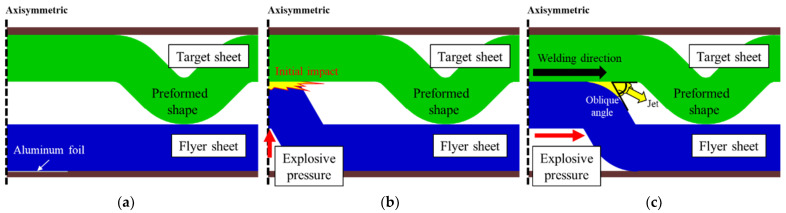
Welding mechanism of the standoff-free VFAW test: (**a**) initial set-up by applying preform target sheet; (**b**) initial impact; (**c**) continuous welding mechanism after initial impact.

**Figure 8 materials-14-04969-f008:**
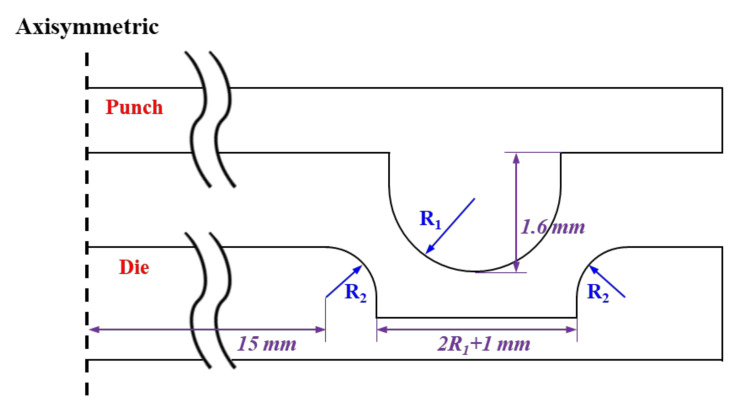
Specification of die set for the preformed target sheet.

**Figure 9 materials-14-04969-f009:**
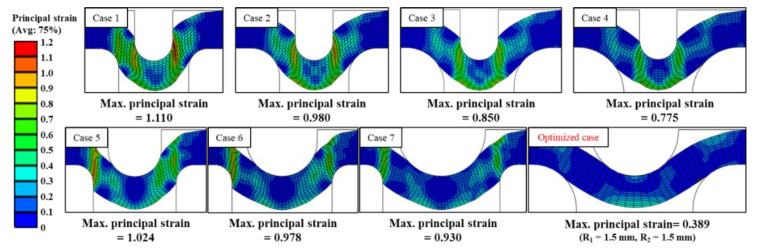
Simulation results for design variables in preforming process for the target sheet.

**Figure 10 materials-14-04969-f010:**
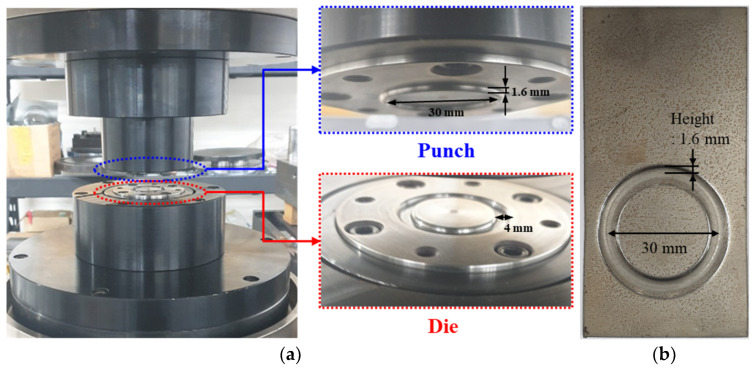
Preform die set: (**a**) punch and die set; (**b**) preformed target sheet.

**Figure 11 materials-14-04969-f011:**
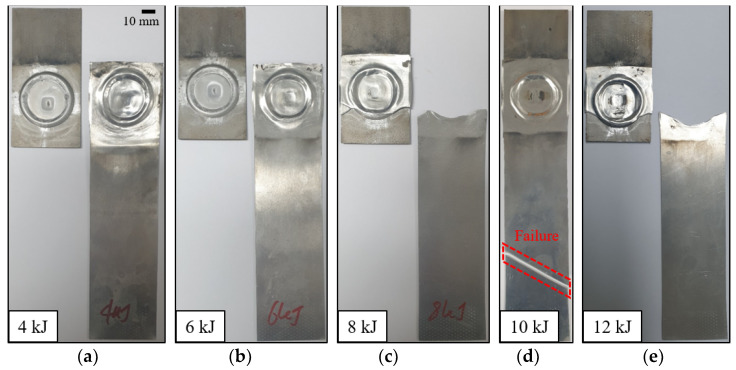
Test specimens after lap shear tests with respect to the input energy: (**a**) 4 kJ; (**b**) 6 kJ; (**c**) 8 kJ; (**d**) 10 kJ; (**e**) 12 kJ.

**Figure 12 materials-14-04969-f012:**
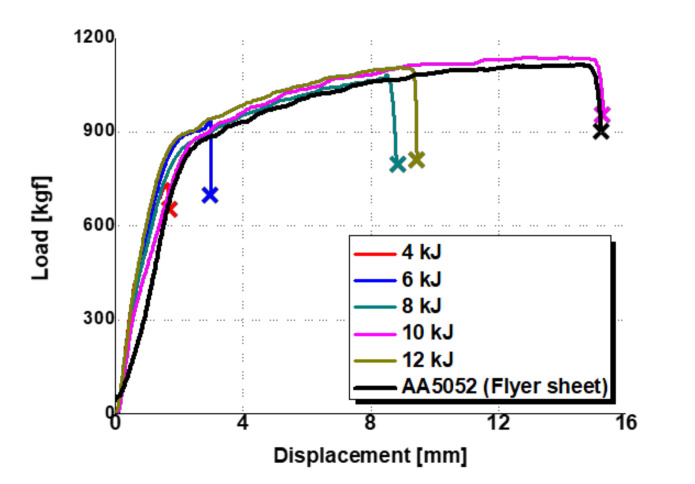
Load–displacement curve in a lap shear test with respect to the input energy.

**Figure 13 materials-14-04969-f013:**
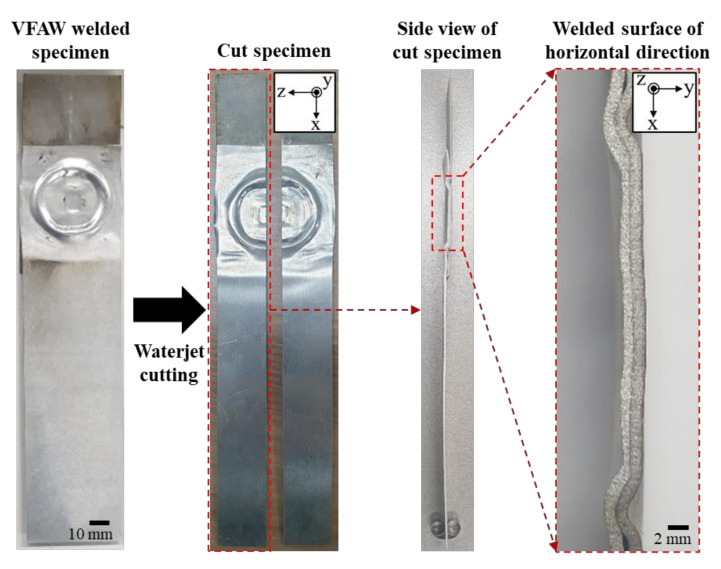
Preparation of the welded specimen from the VFAW test for thickness measurement.

**Figure 14 materials-14-04969-f014:**
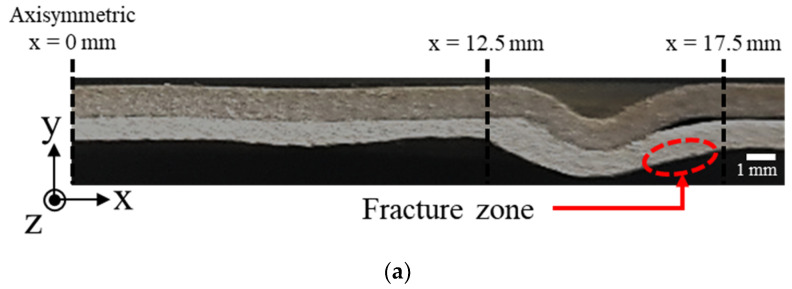
Thickness distribution of welded specimen with increase of input energy: (**a**) sectional view of welded surface; (**b**) thickness distribution along flyer sheet; (**c**) thickness distribution along target sheet.

**Figure 15 materials-14-04969-f015:**
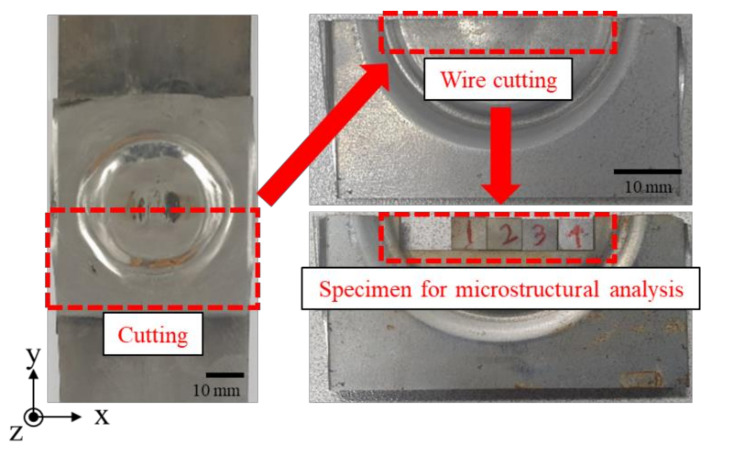
Preparation for microstructural investigation in the welded VFAW specimen.

**Figure 16 materials-14-04969-f016:**
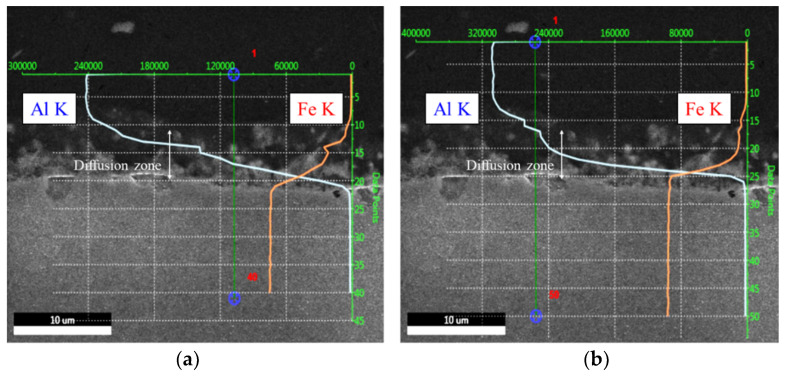
SEM-EDS results about composition at the interface area: (**a**) in specimen-1; (**b**) in specimen-2.

**Figure 17 materials-14-04969-f017:**
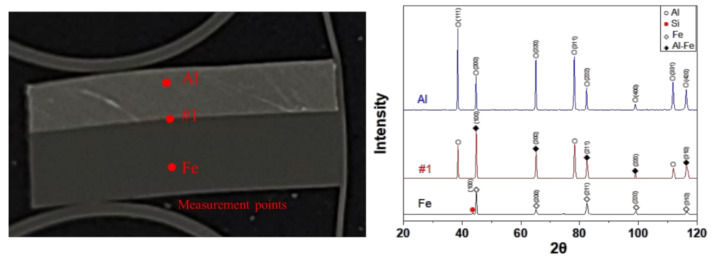
Micro XRD analysis at flyer (Al sheet), interface (#1), and target sheet TRIP1180 sheet).

**Figure 18 materials-14-04969-f018:**
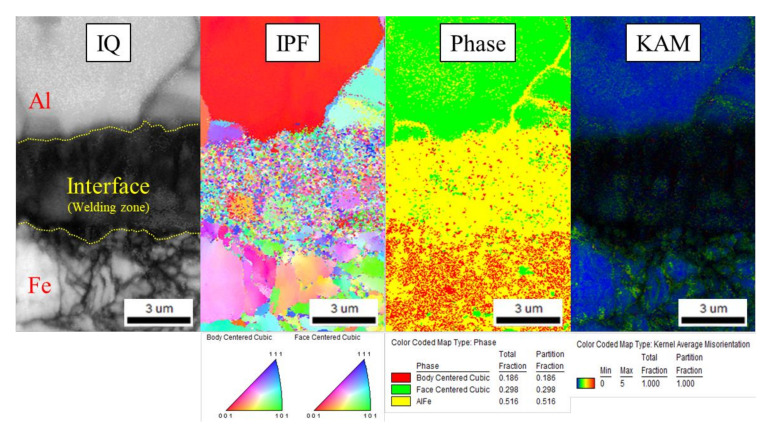
EBSD investigation in the welded VFAW specimen.

**Figure 19 materials-14-04969-f019:**
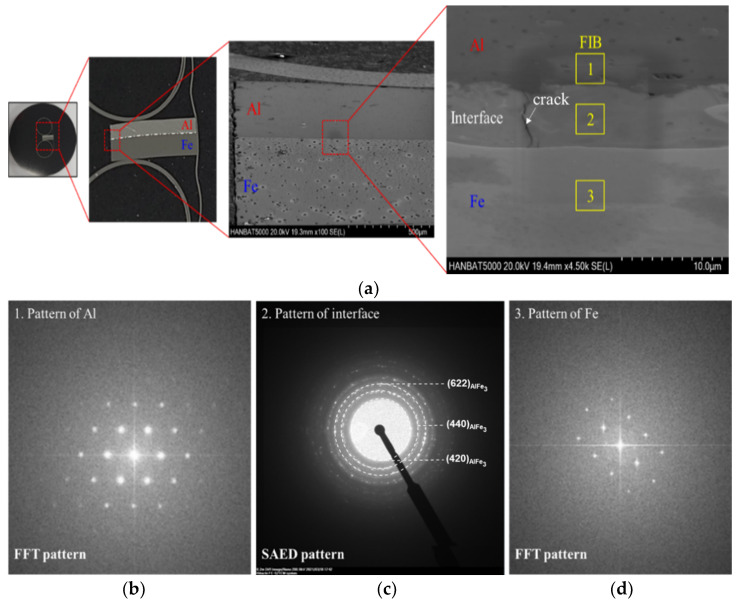
TEM analysis with welded specimen: (**a**) position of TEM specimen; FFT and SAED patterns at (**b**) Al side; (**c**) joint interface; (**d**) Fe side.

**Table 1 materials-14-04969-t001:** Design variables in the preforming process.

Unit: mm	Case 1	Case 2	Case 3	Case 4	Case 5	Case 6	Case 7
Punch radius (R_1_)	0.6	0.6	0.6	0.6	0.9	1.2	1.5
Die radius (R_2_)	0.6	0.9	1.2	1.5	0.6	0.6	0.6

## Data Availability

All the data are available within the manuscript.

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
