# Peer review of "Dissimilar Materials Welding with a Standoff-Free Vaporizing Foil Actuator between TRIP 1180 Steel Sheets and AA5052 Alloy"

_materials, 2021, doi:10.3390/ma14174969_

Round 1
Reviewer 1 Report
The current paper discusses dissimilar materials welding with a standoff-free vaporizing foil actuator between TRIP 1180 steel sheets and AA5052 alloy. Using this technique, a promising application for a light-weight structural materials can be developed and utilized in transportation sector.
The manuscript is well written and structured. However, few comments must be revised before the manuscript get accepted for publication.
1- The abstract is not rich with findings, it is rather summarizing the methods and casting introductory information. Authors are requested to rewrite the abstract.
2- The results need to be improved and authors must elaborate more on the experimental findings. For instance, there is no direct link between the experimental results (such as connections between XRD, EBSD, EDS and TEM results) in terms of phase types, microstructural findings, etc.
3- The TEM results in any manuscript are treasures when they are well explained. However, in this submission, the authors did not give proper explanation to TEM findings.
4- The remaining comments can be found in the attached PDF file.

Reviewer 2 Report
This paper presented a laboratory experiment on vaporizing foil actuator welding (VFAW) of dissimilar alloys. The process parameters were selected based on numerical simulation and validated by experimental observation. The welded surface was examined by SEM-EDS, XRD, EBDS, and TEM. Additionally, lap shear tests were also performed for the selected joint. It was concluded that good quality joint can be achievable by VFAW with input energy of 10 kJ. The results were drawn based on experimental observation and the paper will attract readers in this subject area. However, the following needs to be addressed before publication:
- Abstract, Line 13 : TRIP1180 steel (1.2t), 1.2 t is not an internationally known parameter. Please provide details of the meaning of 1.2 t and 1.0 t.
- Abstract, Line 14: Please use abbreviations after the full form, not before the full form such as vaporizing foil actuator welding (VFAW).
- Introduction, Line 29-31: To increase the readership in the broad area of dissimilar metal joining, please include other joining techniques such as self-piercing riveting and clinching and cite the following two references:
Quality of self-piercing riveting (SPR) joints from cross-sectional perspective: A review- Archives of Civil and Mechanical Engineering-Volume 18
Research on the Influence of the AW 5754 Aluminum Alloy State Condition and Sheet Arrangements with AW 6082 Aluminum Alloy on the Forming Process and Strength of the ClinchRivet Joints, Materials 2021, 14(11), 2980 - Figure 5: Please use the same scale bar for both materials.
- Table 1: the parameters are a bit confusing. For example: What is the R2 for case 5? Please present them in a better way.
- Please include scale bars in Figures 11, 13, 14, and 15.
- Please give a better name for section 3.2
- Please provide a hypothetical reason for why the failure occurred in the sheet for 10 kJ. Is it some defect on the sheet?
Reviewer 3 Report
Dear Authors,
The manuscript is based on a wide range of experimental research. The title and the abstract are appropriate for the content of the text. In general, the manuscript is well constructed.
The article presents steel and aluminum alloy which have been welded by utilizing the standoff-free VFAW.
It has been proposed to perform the VFAW test with applying a preformed target sheet instead of utilizing a conventional standoff. To optimize the preform shape, FEM analysis has been carried out, which was confirmed by experiments for perfect welding between steel sheet and aluminum alloy sheet.
The FEM analysis enables perfect welding between steel and aluminum alloy by applying the input energy of 10 kJ.
The results of the research showed that, the driving force of the high weldability between the two materials resulted from the metallic bonding, observed in the microstructural analysis.
The paper has a major interest for scientific community as it merges applicational and scientific aspects.
The paper is well written and well presented. However, some improvements can be done:
- Tables and Schemes are explained in the two separate paragraphs in the section “ Preform design of target sheet and Figures”. Please, regroup them in one paragraph.
- It is necessary to explain the details of shear tests in the “ Experimental procedure”
- It would be valuable to show the photographs of the samples after the shear test.
Reviewer 4 Report
The author carried out comprehensive studies of the processes occurring during welding of steel TRIP1180 (1.2 t) and AA5052-H32 (1.0 t) by the VFAW method, and also studied the microstructure and phase composition of the weld.
However, it is necessary to highlight a few notes:
1) In the course of determining the phase composition of the metal at the interface (Fig. 19, c), the presence of the FeAl2 phase is shown. Since the intermetallic compounds FeAl3, Fe2Al5, and Fe3Al stand out among the intermetallic compounds with the highest Gibbs energy in the Al-Fe system, it is necessary to explain the presence of the intermetallic compound FeAl2 at the interface.
For explanation, it is recommended to study and refer to the following works:
- Kuz’min M.P., Ivanov N.A., Kondrat’ev V.V., Kuz’mina M.Yu., Begunov A.I., Kuz’mina A.S., Ivanchik N.N. Preparation of aluminum–carbon nanotubes composite material by hot pressing // Metallurgist. – 2018. – Vol. 61 – P. 815–821.
- Kuz'min M.P., Paul K. Chu, Abdul M. Qasim, Larionov L.M., Kuz'mina M.Yu., Kuz’min P.B. Obtaining of Al–Si foundry alloys using amorphous microsilica – Crystalline silicon production waste // Journal of Alloys and Compounds. – 2019. – V. 806 – P. 806–813.
- Kuz’min M.P., Xiao-Yuan Li, Kuz’mina M.Y., Begunov A.I., Zhuravlyova A.S. Changing the properties of indium tin oxide by introducing aluminum cations // Electrochemistry Communications. – 2016. – V. 67 – P. 35–38.
2) Also on line 256 the author uses the term "complex phase", which is not entirely correct. It is recommended to use the concept of intermetallic phase.
I believe that the article is made at a high level and can be published after correcting the noted remarks.
Round 2
Reviewer 2 Report
The authors made a significant effort to improve the manuscript. A minor comment:
Please write down all the individual values in Table 1. What is the value for R2 in case 5? Is it 1.5? or is it 0.6? A bit confusing.
